# Efficient High-Dimensional Quantum Key Distribution with Hybrid Encoding

**DOI:** 10.3390/e21010080

**Published:** 2019-01-17

**Authors:** Yonggi Jo, Hee Su Park, Seung-Woo Lee, Wonmin Son

**Affiliations:** 1Department of Physics, Sogang University, Seoul 04107, Korea; 2Research Institute for Basic Science, Sogang University, Seoul 04107, Korea; 3Korea Research Institute of Standards and Science, Daejeon 34113, Korea; 4Quantum Universe Center, Korea Institute for Advanced Study, Seoul 02455, Korea

**Keywords:** quantum cryptography, quantum key distribution, high-dimensional quantum states

## Abstract

We propose a schematic setup of quantum key distribution (QKD) with an improved secret key rate based on high-dimensional quantum states. Two degrees-of-freedom of a single photon, orbital angular momentum modes, and multi-path modes, are used to encode secret key information. Its practical implementation consists of optical elements that are within the reach of current technologies such as a multiport interferometer. We show that the proposed feasible protocol has improved the secret key rate with much sophistication compared to the previous 2-dimensional protocol known as the detector-device-independent QKD.

## 1. Introduction

Quantum key distribution (QKD) is a novel scheme to distribute a symmetric secret key between two distant authorized parties, Alice, and Bob, by exploiting quantum mechanical phenomena. The protocol provides an information-theoretic security under potential attacks of a malicious eavesdropper, conventionally called Eve. Since its first seminal proposal called BB84 protocol [1], many relevant or extended versions of QKD protocols have been proposed and studied based on quantum principles [2,3,4,5,6,7,8].

In recent QKD studies, the security defects due to device imperfections have been emerging as an important issue. It has been shown that Eve can hack into the QKD system by exploiting the imperfection of devices. This is known as a side channel attack including photon number splitting (PNS) attack [9], faked-state attack [10], detector efficiency mismatch attack [11], detector blinding attack [12,13], time-shift attack [14], and laser damage attack [15]. In this background, measurement-device-independent QKD (MDI-QKD) was proposed to overcome the problems coming from imperfections of measurement devices [16]. In MDI-QKD, the Bell state measurement (BSM) of two photons [17] is an essential task. However, the success probability of BSM with linear optics on single photons is upper bounded by 50% [18,19]. Recent advanced schemes of BSM require multi-photon encoding of 2-dimensional quantum states, called qubits, to beat the limit with linear optics [20,21,22,23,24]. Then, detector-device-independent QKD (DDI-QKD) was proposed [25,26,27] to simplify the scheme of MDI-QKD, exploiting two different degrees-of-freedoms (DoFs) in a single photon and single-photon interference instead of two-photon interference. In its protocol, Alice encodes her information into one DoF of a single photon and sends it to Bob, who encodes his information into another DoF of the single photon. The measurement result of the single photon reveals correlation of two DoFs in the single photon. The implementation of DDI-QKD requires only measurements on single photons and is thus less challenging than BSM performed on two photons. As its scheme is similar to the process of BSM used in MDI-QKD, it was conjectured that DDI-QKD guarantees the same security level with MDI-QKD. However, it has been shown that not all the side channel attacks are protected with DDI-QKD [28,29], and an assumption of the trusted measurement setup is necessary for ensuring its security.

In another branch of QKD research, there has been significant effort to improve the secret key rate, for example, using *d*-dimensional quantum states, called qudits. There are several advantages to using qudits as a generalized information carrier. For example, qudits (d>2) can naturally carry more classical information than qubits. Compared to qubit operations, qudits has been shown to be more robust against quantum cloning (i.e., a possible eavesdropping) [30,31,32]. It has been also found that the efficiency of key distribution increases with qudits in an ideal situation [32,33,34,35,36]. Various high-dimensional QKD protocols have been proposed such as a generalized version of BB84, a multipartite high-dimensional QKD [37], and MDI-QKDs using high-dimensional quantum states [38,39,40]. Moreover, QKD protocols using qudits have been implemented experimentally in various quantum system, for instance, energy-time eigenstates [41,42,43,44,45] and orbital angular momentum (OAM) mode of a single photon [46,47,48,49,50].

In this article, we propose a schematic configuration of high-dimensional QKD based on hybrid encoding over two different DoFs. We demonstrate that the secret key rate is improved with our scheme over previous 2-dimensional QKD based on two different DoFs of a single photon. We also present its implementation with current optical technologies, by exploiting the OAM mode of a single photon as a high-dimensional information carrier. We evaluate the secret key rate of our scheme with respect to the experimental parameters and identify the regime where our scheme is more secure than the original DDI-QKD. In addition, we also compare the security of our scheme with that of high-dimensional MDI-QKD (for the case of d=3).

We note that our protocol is more secure than the original BB84 protocol against a side channel attack (but less secure than MDI-QKD). For example, it can detect the basic detector blinding attack [12] from double clicks of detectors [28,29]. On the other hand, the attained key rate with our protocol is comparable with BB84 protocol, while MDI-QKD has a half of signal sifting rate of BB84 protocol due to the 50% limit of the success probability of the BSM. Although DDI-QKD is not as secure as MDI-QKD and requires trusted elements in BSM setup, the main idea of employing two different DoFs motivated by DDI-QKD still merits consideration for practical usage in some secure communications, e.g., quantum secret sharing [51]. As we demonstrate in this article, it is possible to improve the security as well as the efficiency over the original DDI-QKD in high-dimensional approach. In addition, we here propose a feasible high-dimensional QKD scheme with OAM of a single photon, while a high-dimensional MDI-QKD may be hard to realize due to the difficulty in implementing high-dimensional BSM on two photons with linear optical elements [19].

This article is organized as follows. A schematic description of the *d*-dimensional QKD (*d*-QKD) with hybrid encoding is presented in Section 2, and its practical implementation is in Section 3. In Section 4, we analyze the secure key rate of our protocol. Finally, conclusion on the efficiencies is drawn in Section 5.

## 2. Schematic Description

In this section, we describe a schematic setup of *d*-QKD with hybrid encoding. As an example, the schematic setup of 3-dimensional QKD (3*d*-QKD) with hybrid encoding is shown in Figure 1. In *d*-QKD with hybrid encoding, we exploit OAM mode and multipath mode of a single photon as a information carrier, since the OAM mode is known to be suitable for quantum communication as it is resilient against perturbation effects [52].

As a first step of the protocol, Alice generates *d*-dimensional information randomly. Subsequently, Alice randomly chooses a encoding basis between two mutually unbiased bases (MUBs) which are written as $\left\{|l_{x}\rangle \right\}$ and $\left\{|{\bar{l}}_{x}\rangle \right\}$ where x∈{0,1,2,…,d−1}. The relation between the two MUBs is described as the *d*-dimensional discrete Fourier transformation on the *d* OAM modes which is shown in Equation (Equation 1):
(1)$$\begin{array}{cc} |{\bar{l}}_{x}\rangle & =\frac{1}{d}\sum_{k=0}^{d-1}\omega^{xk}|l_{k}\rangle \end{array}$$
where $\omega =\exp\left(2\pi i/d\right)$. Alice encodes her *d*-dimensional information in OAM modes of a single photon [53]. For example, when d=3, Alice’s classical information $x_{3}$ would be one of the dimensional integers, $x_{3}\in \left\{0,1,2\right\}$, and she generates a quantum state denoted as $|l_{x_{3}}\rangle$ whose OAM value is $\left(x_{3}-1\right)$.

Subsequently, Alice sends the encoded photon to Bob, who encodes his *d*-dimensional information in multipath modes of the single photon. Bob also uses two MUBs that are described as $\left\{|p_{y}\rangle \right\}$ and $\left\{|{\bar{p}}_{y}\rangle \right\}$. $|p_{y}\rangle$ denotes a single photon state in the optical path $p_{y}$ where y∈{0,1,2,…,d−1}. Similarly with Alice’s bases, the relation between Bob’s two bases is given as the *d*-dimensional discrete Fourier transformation of the *d* path modes. Figure 2 shows a schematic setup of Bob’s encoding systems. Bob randomly chooses one basis between path modes and bar path modes, which are MUBs of the path modes. If Bob uses a path mode, he selects one optical path among $\left\{p_{0},p_{1},p_{2}\right\}$ corresponding to his information. If he chooses a bar path mode, he encodes his information by selecting a phases set {B1,B2} in Figure 2b among {1,1}, $\left\{\omega ,\omega^{2}\right\}$, and $\left\{\omega^{2},\omega \right\}$.

After Bob’s encoding, the two qudits encoded in the single photon, which can be written as $|l_{x},p_{y}\rangle$, go into a cyclic transformation of OAM modes. A transformed OAM value of the single photon in path $p_{y}$ is obtained with the following rule: x→x+d−y (mod *d*). Subsequently, a single photon interference is performed by recombining *d*-path via beam splitters that have different transmissivity. The unitary transformation on path modes operated by multi-port interferometer, called tritter [54], is defined as the *d*-dimensional discrete Fourier transformation on the *d* path modes as shown in Equation (Equation 2):
(2)$$\begin{array}{c} {\hat{U}}_{d}=\frac{1}{\sqrt{d}}\left(\begin{array}{cccccc} 1 & 1 & 1 & 1 & \cdots & 1\\ 1 & \omega & \omega^{2} & \omega^{3} & \cdots & \omega^{d-1}\\ 1 & \omega^{2} & \omega^{4} & \omega^{6} & \cdots & \omega^{2(d-1)}\\ 1 & \omega^{3} & \omega^{6} & \omega^{9} & \cdots & \omega^{3(d-1)}\\ \vdots & \vdots & \vdots & \vdots & \ddots & \vdots \\ 1 & \omega^{d-1} & \omega^{2(d-1)} & \omega^{3(d-1)} & \cdots & \omega^{\left(d-1)(d-1\right)} \end{array}\right). \end{array}$$


Subsequently, a OAM value of the single photon is measured in each output port of the tritter. The result of the measurement is obtained from click of a single photon detector. A click in one of the $d^{2}$ detectors corresponds to a projection into one of the following two qudits encoded in a single photon written in Equation (Equation 3):
(3)$$\begin{array}{c} |\Phi_{di+j}\rangle =\frac{1}{\sqrt{d}}\sum_{x=0}^{d-1}\omega^{jx}|l_{x},p_{x+i}\rangle , \end{array}$$
where i,j∈{0,1,2,…,d−1}, and (mod *d*) is omitted in the subscript of *p*. Since the states have the similar form to the *d*-dimensional Bell states, it is expected that the states can be used to distribute a secret key between Alice and Bob. The relation between two qudits encoded in a single photon that enters into the tritter and its corresponding detector click event is shown in Equation (Equation 4):
(4)$$\begin{array}{c} |\Phi_{di+j}\rangle \to D(l_{d-i},p_{d-j}) \end{array}$$
where i,j∈{0,1,2,…,d−1}, and we label a click event of a single photon detector as $D(l_{x},p_{y})$ corresponding to the single photon whose OAM value is $l_{x}$ and path mode is $p_{y}$ after the tritter operation. Since there are $d^{2}$ orthonormal states in Equation (Equation 3), the measurement setup should include $d^{2}$ single photon detectors for one-to-one correspondence of the states and the detectors.

In order to share a secret key, it is necessary to retrieve Bob’s information based on the basis choice of Alice and Bob, and the result of the measurement. For restoration, Alice sends her basis choice to Bob. The method of the restoration is shown in Table 1 as an example when 3*d*-QKD with hybrid encoding is performed. Bob announces only the basis matching information through classical communication, not the result of the measurement. Alice does not need to know the measurement outcome, since Bob already retrieved his encoded information by using the result.

## 3. Experimental Implementation

We investigate a practical implementation of experimental elements that can construct *d*-QKD with hybrid encoding. Alice can generate a single photon OAM state by means of a spatial light modulator (SLM) [55]. SLMs usually have a limited frame rate of around 60 Hz, for fast generation of various OAM values, a digital micromirror device(DMD) is more desirable [56]. An OAM sorter based on liquid crystal devices can also generate photonic OAM states [57,58].

Bob’s path encoding system is realizable with an optical switch over *d*-port, and a schematic setup is shown in Figure 2a. Bob’s bar path encoding system can be changed from Figure 2b by using an optical *d*-port switch and a *d*-port tritter, whose operation on path modes is the *d*-dimensional discrete Fourier transformation as shown in Equation (Equation 2). With the tritter, Bob can choose bar path mode by selecting an input port of the tritter rather than controlling the phase shifters in Figure 2b.

After Bob’s encoding, cyclic transformations of OAM modes are performed in the each port. Figure 3 shows a schematic setup of three-fold OAM cyclic transformation (+1) of OAM values {−1,0,1}. The setup consists of OAM holograms, mirrors, beam splitters and OAM beam splitters (OAM BSs). An OAM BS, composed of a Mach-Zehnder interferometer with a Dove prism in each arm, sorts individual photons based on their OAM value [59]. α is defined from relative angle α/2 between the two Dove prisms and relative phase between photons in the two arms is given by exp(ilα). The three-fold OAM cyclic transformation (+1) consists of three OAM BSs whose α are π, π/2, and −π. The first OAM BS (α=π) and the final OAM BS (α=−π) change the direction of propagation of a photon whose OAM value is odd and even, respectively. The second OAM BS (α=π/2) spatially separates photons whose OAM value is 0 and 2. Photons are separated and combined spatially by using the OAM BSs according to their OAM value. With OAM holograms on each arm, the three-fold cyclic transformation of OAM modes {−1,0,1} is accomplished as it is shown in Figure 3. The experimental setup of the four-fold and five-fold cyclic transformation of OAM modes were proposed and demonstrated as well [60,61,62,63]. While theoretical efficiency of the four-fold cyclic transformation is 100%, fidelity of 4-dimensional Bell state transformation using the four-fold cyclic transformation setup was reported as roughly 91.5% due to reflectivity of optical elements and misalignment [61].

Subsequently, *d*-port single photon interference is performed by using the tritter shown in Equation (Equation 2). The tritter can be implemented with only linear optical elements which are beam splitters, mirrors, and phase shifters. After the interference, an OAM value of the single photon is measured. Direct measurements of an OAM value of a single photon have been studied recently, for instance, by using refractive optical elements that convert OAM modes into transverse momentum modes [64,65], refractive optical elements that give spatial separation of OAM modes [66], sequential weak and strong measurements [67,68], spectrum analysis based on the rotational Doppler effect [69], and interferogram analysis with a multipixel camera [70].

There has been an experimental demonstration of the prepare-and-measure qudit QKD using seven OAM values of a single photon, which includes DMD for fast generation of single photon OAM states and spatial separation of OAM modes proposed in for OAM mode detection [56,66]. In the experiment, it was reported that the efficiency of OAM mode separation was 93%. It is expected that an experimental demonstration of *d*-QKD with hybrid encoding is possible by using above technologies as well as the prepare-and-measure qudit QKD.

## 4. Security Analysis

Before we analyze security of *d*-QKD with hybrid encoding, we need to assume constraints to construct secure *d*-QKD with hybrid encoding as it is studied in [28]: (i) Alice’s and Bob’s random number generators and their classical post-processing should be trusted. (ii) Alice’s and Bob’s encoding systems should be fully characterized and not be influenced by Eve. (iii) Eve cannot physically access to the output ports of the interferometer, in our protocol, the tritter. (iv) The detectors may have some imperfections, but the defects is not from Eve. The first assumption is essential for all QKD schemes to ensure security. The first and second assumptions are necessary for MDI-QKD as well. The third and final assumptions are different from the scenario of MDI-QKD. They are necessary to prevent particular classes of side channel attacks [28,29]. The third assumption can be considered not impractical, since *d*-QKD with hybrid encoding has the similar experimental situation to prepare-and-measure QKD protocols like original BB84. In the situation, Bob can have full measurement setup in his room and he can block access from the outside.

Let us consider several side channel attacks against *d*-QKD with hybrid encoding. Since its similarity of principles to the original DDI-QKD, the security of *d*-QKD with hybrid encoding against side channel attacks is comparable to that of the original DDI-QKD studied in [28,29]. Faked-state attack [10], detector efficiency mismatch attack [11], and time-shift attack [14] are not compatible with assumption (iv) since the attacks require a prior knowledge of imperfections of the single photon detectors. Trojan-horse attack based on back reflection [71,72,73] is considerable. In Trojan-horse attack based on back reflection, Eve sends multi-photon states into Alice’s(Bob’s) encoding system. The photons are reflected at the elements in the encoding system. Then Eve can obtain information about a generated single photon state by analyzing the reflected beam. The attack can be prevented by using frequency filters and isolators like in MDI-QKD case. Trojan-horse attack proposed in [74] is forbidden by assumption (iv), since the detectors in the measurement setup should be manufactured by Eve to accomplish this attack.

Detector blinding attack can threaten QKD systems as well. An essential procedure of the detector blinding attack is that Eve shines strong classical light onto detectors, avalanche photodiodes, to change their mode from Geiger mode to linear mode [12]. In the linear mode, a detection signal can be generated by the strong light pulse that exceeds a threshold. This means that Eve can control a detector click by regulating amplitude of the light pulse. If a threshold of the all detectors is identical, the basic detector blinding attack can be detected by Bob. Let us define the threshold μ. Then the amplitude of Eve’s light pulse should be larger than μ when it arrives at detectors. Eve intercepts Alice’s signal and resends a strong light corresponding to the measured quantum state. When Eve’s and Bob’s bases are matched, for example OAM modes and path modes, the amplitude of Eve’s light pulse should be larger than dμ to make a detector click since the tritter splits the light pulse into four output ports identically. In the situation, Bob can notice the detector blinding attack since *d* detectors are clicked simultaneously. Bob can make an error rate be affected by the attack by assigning random number when more than two detectors are clicked. If there are differences among the threshold of detectors, it is possible that Eve generates one detector click. The clicked detector must have the lowest threshold among the detectors. This means that Eve cannot generate a click of the other detectors independently. So the attack can be found by analyzing statistics of detector clicks.

Detector blinding attack with various blinding power [13] can threaten *d*-QKD with hybrid encoding as well as the original DDI-QKD [29]. However, since the attack requires a prior knowledge about the detectors, it is not compatible with assumption (iv). So we can conclude that without the assumptions which are not necessary in MDI-QKD, the security of *d*-QKD with hybrid encoding cannot be guaranteed against all detector side channel attacks.

To detect Eve’s side channel attacks, we introduce a random tritter operation of Bob. The tritter operation written in Equation (Equation 2) is performed on path modes after Bob’s encoding. It is possible that Bob chooses one of tritter operation among *d* different operations ratter than a fixed operation. For example, Bob can chooses one operation among the operations shown in Equation (Equation 5):
(5)$$\begin{array}{cccc} {\hat{U}}_{3,0} & =\frac{1}{\sqrt{3}}\left(\begin{array}{ccc} 1 & 1 & 1\\ 1 & \omega & \omega^{2}\\ 1 & \omega^{2} & \omega \end{array}\right),\; & {\hat{U}}_{3,1}=\frac{1}{\sqrt{3}}\left(\begin{array}{ccc} \omega^{2} & 1 & \omega \\ 1 & 1 & 1\\ \omega & 1 & \omega^{2} \end{array}\right),\;{\hat{U}}_{3,2} & =\frac{1}{\sqrt{3}}\left(\begin{array}{ccc} \omega^{2} & \omega & 1\\ \omega & \omega^{2} & 1\\ 1 & 1 & 1 \end{array}\right) \end{array}$$
for 3*d*-QKD with hybrid encoding. The operations can be implemented by using 3*d*-tritter and phase shifters.

Let us consider the case that Bob chooses path mode $|{\bar{p}}_{0}\rangle$ and Eve tries detector blinding attack with strong pulse whose OAM mode is $|{\bar{l}}_{0}\rangle$. If Bob chooses *t*, where t∈{0,1,2}, and performs the tritter operation ${\hat{U}}_{3,t}$, the pulse goes to output port $p_{t}$ of the tritter. For a successful attack, Eve must find the pulse intensity and that one detector in the output port is clicked and the other detectors are not, regardless of Bob’s choice of tritter operation. Also, Eve should perform detector blinding attacks with various blinding power and find at least three different blinding powers, since Bob monitors statistics of outcomes. For instance, Bob can check whether $|{\bar{l}}_{0},{\bar{p}}_{0}\rangle$ is projected onto $|\Phi_{0}\rangle$, $|\Phi_{3}\rangle$, and $|\Phi_{6}\rangle$ equally or not. Therefore, Eve should prepare at least three different pulse intensities, which occur click events of different detectors on the same output port, to pass Bob’s statistics check.

For a large dimension, we expect that such attack is improbable with the assumption (iv), i.e., with trusted devices. Since click thresholds of different detectors are very similar but randomly fluctuated, it is difficult to find blinding powers and pulse intensities that satisfy the successful attack conditions. For a successful attack, Eve should find the powers and intensities that only one detector is clicked while the other d−1 detectors in the port are not clicked, and the attack does not influence Bob’s outcome statistics regardless of Bob’s choice of tritter operation ${\hat{U}}_{d,t}$, where t∈{0,1,2,…,d−1}. Therefore, we expect that side channel attacks are probably detected for a high-dimensional QKD using hybrid encoding, if Bob applies the random choice of tritter operation and a countermeasure of a side channel attack, such as the random-detector-efficiency protocol [75,76]. Compared to this, prepare-and-measure QKD protocols using high-dimensional systems are threatened by the first proposal of detector blinding attack [12], and the original DDI-QKD was breached by the combined attack of the detector blinding attack with various blinding power and detector efficiency mismatched attacks even with the random-detector-efficiency protocol [13]. Therefore, we can conclude that the complexity of a successful side channel attack becomes higher by exploiting the proposed protocol compared to prepare-and-measure *d*-QKDs and the original DDI-QKD, although the proposed protocol does not provide the detector-device-independent security.

It is necessary to analyze security of *d*-QKD with hybrid encoding to evaluate the usefulness of the protocol. The analysis of the security is able to be made through the inspection of the equivalent protocol using the entanglement distillation process (EDP) [4,5,6]. The idea of the method is that, if Alice and Bob share the maximally entangled state, Eve cannot generate correlation between her state and the shared maximally entangled state of Alice and Bob [77]. In the method, we can analyze the security of the proposed protocol with the amount of distributed maximally entangled states. In order to use the method, an equivalent protocol of which Alice and Bob share an entangled state at the end should be introduced. Note that the equivalent protocol is employed only for the security analysis, so its experimental efficiency is not significant. However, it is important that the equivalent protocol is physically realizable, since any security analysis of QKD should be valid under quantum mechanics. Therefore, we will briefly introduce possible implementations of the equivalent protocol to show that it is physically reasonable.

At first, Alice and Bob generate the three-photon entangled state shown in Equation (Equation 6):
(6)$$\begin{array}{c} {|\Psi \rangle }_{ABD}=\frac{1}{d}\sum_{m,n=0}^{d-1}{|l_{m}\rangle }_{A}{|l=0,p_{n}\rangle }_{B}{|l_{m},p_{n}\rangle }_{D}, \end{array}$$
where the subscript *A*(*B*) means Alice’s (Bob’s) single photon state and the subscript *D* means a single photon that goes to tritter and OAM measurement setup. Generation of this state is possible, in principle, by using two cascade spontaneous parametric down-conversion (SPDC) crystals, spatial discrimination elements of the OAM mode, and relabelling of the OAM and path values. For a 4-dimensional system, the generation of 4-dimensional OAM mode entangled states [78] and 4-dimensional path mode entangled states [79,80] using SPDC crystals was demonstrated. Alice and Bob keep their photons, and Bob measures the photon *D* using the measurement setup. Based on the result, Bob performs the corresponding unitary operation to share the maximally entangled state shown in Equation (Equation 7):
(7)$$\begin{array}{c} \frac{1}{\sqrt{d}}\sum_{k=0}^{d-1}{|l_{k}\rangle }_{A}⊗{|p_{k}\rangle }_{B}. \end{array}$$


Alice (Bob) chooses her (his) measurement basis randomly between OAM (path) modes and bar OAM (path) modes. After the measurement, Alice and Bob share their measurement bases and discard if the two bases are not matched. If the two bases are matched, their measurement outcomes are always identical if there is no error and no Eve.

Since the maximally entangled state is distributed to Alice and Bob, security of the protocol becomes the same with that of a *d*-dimensional entanglement based QKD. Security of a QKD using *d*-dimensional maximally entangled states was studied against individual attacks [33] (Eve monitors state separately), and against collective attacks [34,35] (Eve monitors several states jointly). According to the results, secret key rate of QKD using *d*-dimensional quantum states against collective attack is evaluated as shown in Equation (Equation 8):
(8)$$\begin{array}{c} r=\log_{2}d+2Q\log_{2}\left(\frac{Q}{d-1}\right)+\left(1-Q\right)\log_{2}\left(1-Q\right). \end{array}$$


The unit of the secret key rate is (bits/sifted signal). *Q* is state error rate obtained from Equation (Equation 9):
(9)$$\begin{array}{c} Q=\sum_{i\neq j}\langle l_{i},p_{j}|\rho |l_{i},p_{j}\rangle , \end{array}$$
where ρ is the density matrix of the state shared by Alice and Bob, and i,j∈{0,1,2,…,d−1}. In the ideal case, no error and no Eve, since the distributed state is the state described in Equation (Equation 7), the error rate becomes trivial, Q=0.

Now, we investigate an improvement of a secret key rate of *d*-QKD with hybrid encoding compared with the original DDI-QKD. Secret key rates per sifted signal, *r*, of *d*-QKD with hybrid encoding are plotted in Figure 4. Figure 4a shows the secret key rate of the original DDI-QKD (black dotted line), 3*d*- (red dashed line), 4*d*- (blue dot-dashed line), and 5*d*-QKD with hybrid encoding (orange solid line) in the ideal situation. QKD with hybrid encoding using higher dimensional quantum states has a higher secret key rate than the original DDI-QKD at same error rate, since a quantum system in high-dimension can carry more information per single quanta and qudit has enhanced robustness against an optimal cloning and eavesdropping.

In Figure 4b, we simulate secret key rates of *d*-QKD with hybrid encoding and the original DDI-QKD with a change of the realistic experimental factors, transmission loss η and dark count rate of single photon detectors. When a photon propagates through an optical fiber or atmosphere, there is transmission loss. So transmission efficiency is approximately proportional to the distance between Alice and Bob that QKD is able to be achieved. For a single photon detector, since it is very sensitive in order to detect a very weak pulse, a single photon, it is possible to be clicked by background noise even if there is no received photon. The event is called dark count. If there is no Eve, the probability of the detector click corresponding to the state $|\Phi_{0}\rangle$ when Alice encodes *x* and Bob encodes *y* in a single photon is able to be described as follows:
(10)$$\begin{array}{cc} p(x,x) & =\frac{1}{d}\left(1-\eta \right){\left(1-\nu \right)}^{\left(d^{2}-1\right)}+\eta \nu {\left(1-\nu \right)}^{\left(d^{2}-1\right)} \end{array}$$
(11)$$\begin{array}{cc} p(x,y) & =\eta \nu {\left(1-\nu \right)}^{\left(d^{2}-1\right)} \end{array}$$
where x,y∈{0,1,2,…,d−1}, x≠y, *d* is the dimension of quantum states used in *d*-QKD with hybrid encoding, and ν is the dark count rate per pulse. The first term in Equation (Equation 10) denotes the case when the single photon arrives at a detector and it triggers off the detector, while there is no dark count in the other detectors. The second term in Equation (Equation 10) denotes that the single photon detector is clicked due to the dark count when the single photon is lost in channel and the other detectors are not clicked. In the ideal case, no Eve and no state error, p(x,y) should be zero since the state cannot be projected on $|\Phi_{0}\rangle$. The only case that the detector is clicked is that the single photon is lost and the detector is clicked due to the dark count. The error rate in this situation is evaluated from the equation described as follows:
(12)$$\begin{array}{c} Q=\frac{\sum_{i\neq j}p\left(i,j\right)}{\sum_{x,y=0}^{d-1}p\left(x,y\right)}, \end{array}$$
where i,j∈{0,1,2,…,d−1}. The dark count rate, ν, is assumed as $10^{-5}$ per pulse in Figure 4b. In the plot, it is shown that a secret key rate becomes higher, as the dimension of quantum states used in *d*-QKD with hybrid encoding increases in low transmission loss regime. When the transmission loss is high, the secret key rate decreases more rapidly as *d* increases. QKD with hybrid encoding using higher dimensional quantum states is more influenced by the dark count of detectors, since the number of the single photon detector used in *d*-QKD with hybrid encoding is lager than the original DDI-QKD. Therefore, when a single photon is lost, the error rate of QKD with hybrid encoding using higher dimensional quantum states increases rapidly compared with that of the original DDI-QKD.

Now, we compare 3*d*-QKD with hybrid encoding with MDI-QKD using 3-dimensional quantum states (3*d*-MDI-QKD). 3*d*-MDI-QKD was proposed to increase a secret key rate of original MDI-QKD [38]. In its key rate analysis, it is assumed that 3-dimensional BSM used in 3*d*-MDI-QKD includes six single photon detectors and the 3-dimensional BSM setup can discriminate only three 3-dimensional Bell states among nine ones. Figure 5 shows the secret key rate of 3*d*-MDI-QKD (red dashed line) and 3*d*-QKD with hybrid encoding (black solid line). Secret key rate per total pulse can be obtained from (signal sifting rate) × (secret key rate per sifted key *r*). The signal sifting rate is obtained from (the probability that Alice and Bob used the same bases) in *d*-QKD with hybrid encoding, and (the probability that Alice and Bob used the same bases) × (the success probability of a BSM) in MDI-QKD. Since a success probability of BSM with linear optics cannot be 100% [18,19], MDI-QKD always has a lower secret key rate per total signal than prepare-and-measure QKD protocols and QKD with hybrid encoding.

Furthermore, it was proven that a generalized BSM in a high-dimensional two-photon state cannot be implemented by means of linear optical elements [19]. The scheme using multi-photon interference with linear optics can be adopted to implement MDI-QKD using qudits [81], however, the secret key rate *R* of the protocol is always lower than original MDI-QKD, since the signal sifting rate of the protocol is given as $1/(2d^{2})$. There is another scheme in which the ideal signal sifting rate can reach 1/(2d) by exploiting nonlinear effects, however, because of the nonlinearity, experimental efficiency of the scheme is much lower than that of the setup with linear optical elements [82]. Also, it was shown that a secret key rate of MDI-QKD using qudits (d>4) cannot exceed that of qubit MDI-QKD at low error rate even if a signal sifting rate of a *d*-dimensional BSM setup reaches 1/(2d) [82]. Therefore, it can be claimed that QKD with hybrid encoding is more suitable to exploit qudits than MDI-QKD in its implementation, although it needs additional assumptions to guarantee the security level of MDI-QKD.

Here, we compare key generation efficiency of *d*-QKD with hybrid encoding with that of existing *d*-QKDs. First, compared with entanglement-based *d*-QKDs [43,44,45], our protocol has an advantage in that generation of an entangled state is not necessary. A high-dimensional time-energy entangled state is generated from spontaneous parametric down-conversion (SPDC), and entangled state generation efficiency of SPDC is not comparable with a single photon OAM mode encoder.

Key generation efficiency of prepare-and-measure *d*-QKDs [47,48,49,50,83] are comparable with that of our protocol. Our protocol is vulnerable to photon loss noise compared with prepare-and-measure *d*-QKDs, as it is shown in Figure 6. Our protocol employs $d^{2}$ detectors in the measurement setup, while prepare-and-measure *d*-QKDs have 2d detectors. Because of this reason, an effect of a dark count of detectors in our protocol is larger than that in prepare-and-measure *d*-QKDs. This means that growth in an error rate of our protocol is higher than that of prepare-and-measure *d*-QKDs when a single photon is lost. However, as it is shown in the security analysis, our protocol can prevent certain kinds of side channel attacks against detectors, while security of prepare-and-measure *d*-QKDs is threatened even by the first proposal of a detector blinding attack. In consideration of this, the gap between the two secret key rates shown in Figure 6 is not significant.

Finally, we note that it is possible to employ state-of-the-art techniques in our protocol, since our protocol is constructed with general experimental elements. For example, in *d*-QKD using partial MUBs of OAM [49], they proposed using special single photon OAM modes in their protocol for noise robustness. It is expected that the setups used in the protocol are exploited in our protocol for the same purpose as well.

## 5. Conclusions

In this paper, we proposed a schematic configuration of *d*-dimensional QKD based on hybrid encoding over two different DoFs. Qudits are exploited in the setup to improve a secret key rate, since a qudit can carry more classical information and it has enhanced robustness against eavesdropping compared with a qubit. We investigated possible practical implementations of the proposed QKD protocol with current optical technologies. OAM modes of a single photon is exploited as a high-dimensional information carrier. OAM modes are suitable for quantum communication because of their resilience against perturbation effects. We showed that a cyclic transformation of OAM modes can be implemented within the reach of current technologies as well. We analyzed security of the proposed protocol and showed there is improvement compared with original qubit protocol in an ideal situation. We found the condition that *d*-QKD with hybrid encoding has a higher secret key rate than the original DDI-QKD in the consideration of realistic experimental parameters as well. Finally we compared our protocol with existing *d*-QKDs and showed our protocol has advantages regarding the prevention of side channel attacks against detectors and experimental feasibility.

## Figures and Tables

**Figure 1 entropy-21-00080-f001:**
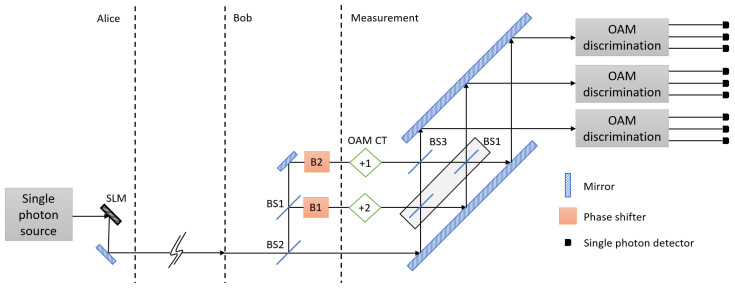
A schematic setup of 3-dimensional quantum key distribution (QKD) with hybrid encoding. Alice uses orbital angular momentum (OAM) modes of a single photon, and Bob controls the phase of each path to encode their information in the single photon. The encoded photon enters into a 3-port interferometer. After single photon interference, a OAM value and existing path of the single photon is measured. SLM: spatial light modulator; BS1: 50:50 beam splitter; BS2: beam splitter of which transmissivity is 1/3; BS3: beam splitter of which transmissivity is 2/3; OAM CT: cyclic transformation of OAM modes.

**Figure 2 entropy-21-00080-f002:**
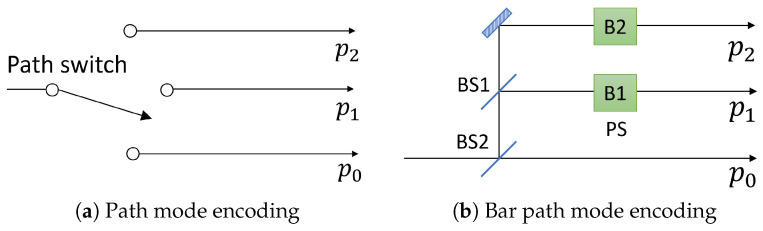
Schematic setups of Bob’s two encoding systems. (**a**) Bob chooses one path to encode his information by using optical switch; (**b**) Bob encodes his information by control phase shifters, B1 and B2. Details are described in the maintext. BS1: 50:50 beam splitter; BS2: beam splitter of which transmissivity is 1/3; PS: phase shifter

**Figure 3 entropy-21-00080-f003:**
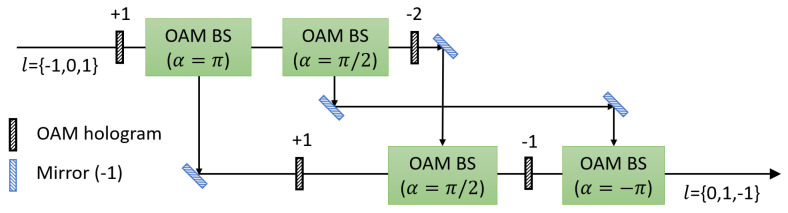
A schematic diagram of experimental setup of three-fold cyclic transformation of OAM modes. There are OAM beam splitters (OAM BSs) which consist of a Mach-Zehnder interferometer with Dove prisms. α/2 means relative angle between the two Dove prisms. The first OAM BS (α=π) and the final OAM BS (α=−π) change a direction of propagation of a photon whose OAM value is odd and even, respectively. The second OAM BS (α=π/2) separates a photon whose OAM value is 0 and 2. With OAM holograms, the three-fold cyclic transformation of OAM modes {−1,0,1} is accomplished.

**Figure 4 entropy-21-00080-f004:**
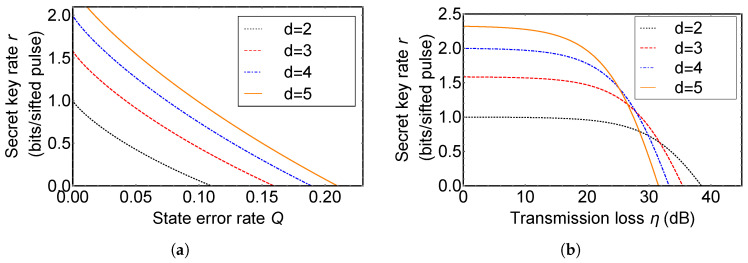
The secret key rate of the original detector-device-independent QKD (DDI-QKD) (black dotted line), 3*d*- (red dashed line), 4*d*- (blue dot-dashed line), and 5*d*-QKD with hybrid encoding (orange solid line). (**a**) Plot of the secret key rate *r* (bits/sifted pulse) vs. state error rate *Q*; (**b**) Plot of the secret key rate *r* (bits/sifted pulse) vs. transmission loss η (dB). Dark count rate of single photon detectors is assumed as $10^{-5}$ per pulse.

**Figure 5 entropy-21-00080-f005:**
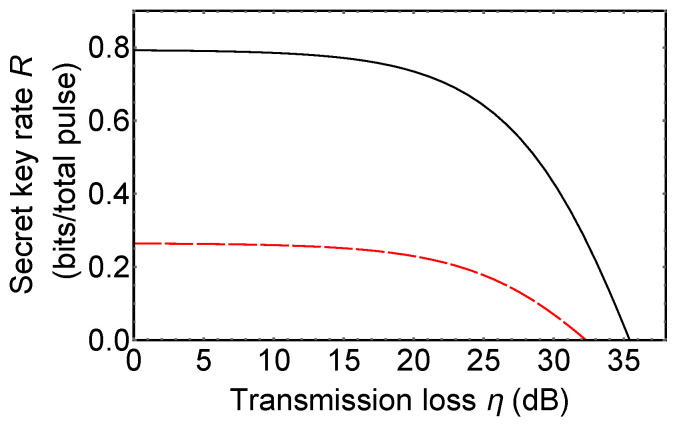
The secret key rate of 3*d*-measurement-device-independent QKD (MDI-QKD) (red dashed line) and 3*d*-QKD with hybrid encoding (black solid line). Plot of the secret key rate *R* (bits/total pulse) vs. transmission loss η (dB). The secret key rate per total signal is obtained from (the secret key rate per sifted key) × (the signal sifting rate). Details are described in maintext. Dark count rate of single photon detectors is assumed as $10^{-5}$ per pulse.

**Figure 6 entropy-21-00080-f006:**
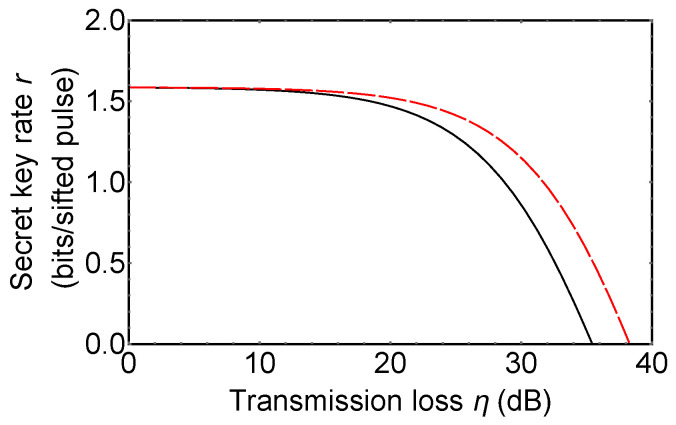
The secret key rate of 3*d*-QKD hybrid encoding (black solid line) and a prepare-and-measure 3*d*-QKD (red dashed line). Dark count rate of single photon detectors is assumed as $10^{-5}$ per pulse.

**Table 1 entropy-21-00080-t001:** An example of Bob’s operation on his encoded information when d=3 and the result of the measurement is $|\Phi_{3i+j}\rangle$. According to their bases choice and the measurement result, it is necessary to retrieve his information for sharing the same information.

Bases	Bob’s Operation ($|\Phi_{3i+j}\rangle$)
bases 1 ($l_{x}$, $p_{y}$)	y→y−i (mod 3)
	1↔2 for j=0
bases 2 (${\bar{l}}_{x}$, ${\bar{p}}_{y}$)	0↔2 for j=1
	0↔1 for j=2

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
