# Peer review of "Efficient High-Dimensional Quantum Key Distribution with Hybrid Encoding"

_entropy, 2019, doi:10.3390/e21010080_

Reviewer 1 Report

In the submitted work entitled “Efficient High-dimensional Quantum Key Distribution with Hybrid Encoding,” the authors claimed that they proposed a schematic setup of quantum key distribution (QKD) with an improved secret key rate based on high-dimensional quantum states by using two degrees-of-freedom of a single photon, orbital angular momentum modes and multi-path modes to encode secret key information. The idea of using degrees-of-freedom to obtain an efficient High-dimensional Quantum Key Distribution is not new and presented by many other authors. I have three main concerns with the paper.

1. The author should compare their proposed method with the current methods in the literature. For example Experimental investigation of high-dimensional quantum key distribution protocols with twisted photons, High-Dimensional Quantum Key Distribution using Dispersive Optics, Towards practical high-speed high dimensional quantum key distribution using partial mutual unbiased basis of photon's orbital angular momentum, High-dimensional quantum key distribution based on multicore fiber using silicon photonic integrated circuits.

2.  The resolution of the figure needs to improve.

3.  The English need considerable improvement. There are some English and Grammatical mistakes. For example in the abstract “has improved secret key” to be “has improved the secret key”. “Two degrees-of-freedom of a single photon, orbital angular momentum modes and multi-path modes, are used to encode secret key information” to be “Two degrees-of-freedom of a single photon, orbital angular momentum modes, and multi-path modes, are used to encode secret key information”.

Author Response

Reply to the first referee,

We would like to reply to the points by the first referee as followings.

Point 1;

> In fact, we had compared our protocol to the entanglement-based d-QKDs including DO(dispersive optics)-QKD and prepare-and-measure d-QKDs. First, our protocol has advantage over entanglement-based d-QKDs in the respect that the efficiency in the generation of an entangled photon pair is not comparable with a single photon OAM encoding. Prepare-and-measure d-QKDs are robust against photon loss noise than our protocol, however, prepare-and-measure d-QKDs is vulnerable to side channel attacks while our protocol can prevent specific kinds of side channel attacks. Also, some state-of-the-art techniques in the d-QKD mentioned in the comment, such as d-QKD using partial mutual unbiased basis of photon’s OAM, can be exploited in our protocol, since our protocol is constructed with general experimental elements.

> The description on these comparisons are added in the manuscript. (Please read the lines 270-288 in the modified manuscript)

> Some references about d-QKD are added for the comparison. (reference number 45, 49, 50, 83)

Point 2;

> The resolution is improved as it is recommended.

Point 3; 

> A great deal of grammatical errors and typos are corrected. The changes are noted in the manuscript.  

Reviewer 2 Report

The authors elaborate on a previous proposal by some of them to present a different protocol for DDI-QKD based on different degrees of freedom of single photons. The analysis is interesting and well motivated within its context. The advantages are discussed in length and clearly.

I can recommend this manuscript for acceptance, if the authors amend some parts that need editing:

in eq. (4) the symbol D is undefined;

the state (6) is actually extremely hard to generate, and, further, the generation rate would likely be low. If this entanglement state is only used for the sake for calculating the attainable key rate, it would be better to shy from comments on its feasibility - this becomes confusing.

Author Response

Related with the second referee's points, we would like to reply as following.

Point 1; 

> Definition of the symbol D is introduced above Eq. (3) in the previous version. For the matter of clarity, the definition is moved to below Eq. (4).

Point 2; 

> The information-theoretic security of BB84 protocol, which is the basic prepare-and-measure QKD protocol, is equivalent to that of E91 protocol, which is identical to the BB84 protocol if we analyze security using entanglement distillation process (EDP). In the case of the security proof, the equivalent protocol is used only for the security analysis. Therefore, experimental realization of the equivalent protocol including an entangled state generation is not necessary to implementable. Similarly, our protocol does not exploit entangled states in its implementation, as it is described in Section 2 and Section 3 and the equivalent protocol, which employ the state described in Eq. (6), is used only for security analysis.

However, we recognize that the description of possible implementations of the equivalent protocol can cause a misunderstanding, so we add the following sentences in the manuscript in order to avoid a possible confusion.

Add: (line 219-223) Note that the equivalent protocol is employed only for the security analysis, so its experimental efficiency is not significant. However, it is important that the equivalent protocol is physically realizable, since any security proof of QKD should be valid under the quantum mechanics. Therefore, we will briefly introduce possible implementations of the equivalent protocol to show that it is physically reasonable.

Round  2

Reviewer 1 Report

The authors did the requested changes.